# Regulation of Cell Adhesion and Migration via Microtubule Cytoskeleton Organization, Cell Polarity, and Phosphoinositide Signaling

**DOI:** 10.3390/biom13101430

**Published:** 2023-09-22

**Authors:** Narendra Thapa, Tianmu Wen, Vincent L. Cryns, Richard A. Anderson

**Affiliations:** 1The Carbone Cancer Center, School of Medicine and Public Health, University of Wisconsin-Madison, 1111 Highland Avenue, Madison, WI 53705, USA; twen4@wisc.edu (T.W.); vlcryns@medicine.wisc.edu (V.L.C.); 2Department of Medicine, School of Medicine and Public Health, University of Wisconsin-Madison, 1111 Highland Avenue, Madison, WI 53705, USA

**Keywords:** phosphoinositide, microtubules, cell polarity, epithelial cell, mesenchymal cells, cell adhesion, cell migration

## Abstract

The capacity for cancer cells to metastasize to distant organs depends on their ability to execute the carefully choreographed processes of cell adhesion and migration. As most human cancers are of epithelial origin (carcinoma), the transcriptional downregulation of adherent/tight junction proteins (e.g., E-cadherin, Claudin and Occludin) with the concomitant gain of adhesive and migratory phenotypes has been extensively studied. Most research and reviews on cell adhesion and migration focus on the actin cytoskeleton and its reorganization. However, metastasizing cancer cells undergo the extensive reorganization of their cytoskeletal system, specifically in originating/nucleation sites of microtubules and their orientation (e.g., from non-centrosomal to centrosomal microtubule organizing centers). The precise mechanisms by which the spatial and temporal reorganization of microtubules are linked functionally with the acquisition of an adhesive and migratory phenotype as epithelial cells reversibly transition into mesenchymal cells during metastasis remains poorly understood. In this Special Issue of “Molecular Mechanisms Underlying Cell Adhesion and Migration”, we highlight cell adhesion and migration from the perspectives of microtubule cytoskeletal reorganization, cell polarity and phosphoinositide signaling.

## 1. Introduction

Cell adhesion and cell migration are integral cellular processes for vertebrates that are crucial for embryonic development and normal cellular functioning [1]. Dysregulated cell adhesion and migration play key roles in tumor metastasis and have remained a prominent focus in cancer research for decades, with numerous excellent reviews on these cancer topics [1,2,3]. From the perspectives of cancer progression and metastasis, most human cancers (85%) are of epithelial origin (carcinoma), in which indolent epithelial cells gain mesenchymal traits via the epithelial–mesenchymal transition (EMT) that imparts adhesive and migratory phenotypes to transformed epithelial cells, enabling distant metastasis [4,5]. Although the loss of E-cadherin-mediated adherent junctions and cell polarity is a well-portrayed hallmark of carcinoma progression, metastasizing cancer cells undergo extensive reorganization of their cytoskeletal system, which encompasses actin, microtubules and intermediate filaments [6]. The dramatic change in spatial orientation and organization, as well as the originating sites of the microtubule cytoskeleton, are some of the most distinguishing features observed in epithelial cells that undergo conversion to mesenchymal cells [7]. Specifically, the non-centrosomal orientation of microtubules at apical membranes in epithelial cells is gradually converted to the centrosomal orientation of these microtubules, and these changes in the organization of the microtubule cytoskeleton coincides temporally with many adhesive and migratory traits acquired by transitioning carcinoma cells [7]. However, changes in microtubule organization/orientation also occur during neuronal polarity and embryonic development [8,9]. Intriguingly, microtubule polymerization and growth inversely regulate focal adhesion assembly [10]; however, how this coordinated interplay between microtubule growth and focal adhesion assembly accomplishes productive cell adhesion and migration is poorly understood. Moreover, there is an emerging link between microtubules, cell polarity and phosphoinositide signaling. Despite microtubules arising as a target for the therapeutic treatment of cancer soon after the start of modern chemotherapy [11], the functional relationship between the therapeutic targeting of microtubules and focal adhesion assembly/focal adhesion signaling has not been well mechanistically defined. In this Special Issue of “Molecular Mechanism Underlying Cell Adhesion and Migration”, we will critically review cell adhesion and migration from the perspectives of microtubule cytoskeletal reorganization. Additionally, we will explore how microtubules collectively regulate cell polarity, phosphoinositide signaling and cell adhesion and migration, which are all key aspects of choreographing metastatic progression.

## 2. Distinct Orientation and Organization of Microtubules in Epithelial vs. Mesenchymal Cells

Microtubules constitute one of the most prominent cytoskeletal elements in mammalian cells and provide structural support as well as a trafficking platform for the transport of intracellular vesicles and organelles [12]. An individual microtubule is a 25-nm-diameter hollow tube made of 13 tubulin filaments, and each filament is a heterodimer of α- and β-tubulin organized in a head-to-tail fashion [12]. This results in a polarized structure of microtubules, with α-tubulin at one end (minus end) and β-tubulin at the other end (plus end) (Figure 1A) extending up to several millimeters in length. The microtubule minus end exhibits slow dynamics, whereas the plus end grows and shrinks rapidly and interacts with different intracellular molecules and structures [12]. Overall, microtubules are highly dynamic structures that undergo repeated cycles of polymerization (growth) and depolymerization (shrinkage) and are finely tuned to the needs of the cells and tissues. 

The intracellular sites from where microtubule nucleation and growth begin are known as microtubule-organizing centers (MTOCs) [13]. One of the distinctive features observed in epithelial and mesenchymal cells is the position of the MTOC and the overall orientation, growth, and organization of the microtubules [13,14]. Polarized epithelial cells contain non-centrosomal MTOCs at apical membrane regions from where microtubules plus ends orient and extend towards basolateral cell membranes (Figure 1B). In contrast, mesenchymal cells contain dominant centrosomal MTOCs in the vicinity of the nucleus and Golgi apparatus, from which a radial array of microtubule plus ends are oriented towards the cell periphery (Figure 1C) [13]. A unique non-centrosomal MTOC at an apical site result in the alignment of long and numerous microtubules along the apical-to-basal axes of polarized epithelial cells. This unique organization of microtubules in epithelial cells was initially discovered in the late 1980s using hook decoration methods and electron microscopy [15]. This arrangement of microtubules allows the sorting of specific membrane components via directing vesicle trafficking to distinct apical and basolateral regions of the plasma membrane [16].

The mechanisms by which MTOC and centrosomes gain or adopt an off-center position toward apical or intercellular junctions in polarized epithelial cells are not precisely understood [17]. They may arise from the microtubule nucleating activity of the γ-tubulin ring complex (γTuRC) near apical regions in epithelial cells [13]. Similarly, they may originate from microtubule release at the centrosomes; microtubule minus ends is subsequently captured and anchored at apical non-centrosomal sites, generating apico-basal arrays of microtubules [6,13]. The analysis of microtubule regrowth in polarized MDCK cells following nocodazole treatment shows the centrosomal nucleation of microtubules and the initial aster formation, which is subsequently replaced by apico-basal microtubule arrays [13]. Centrosome decentering to apical regions may also result from pulling forces from a defined part of the cell periphery [13]. For example, Par3 is present along epithelial cell junctions and is capable of recruiting the dynein motor protein, which pulls on microtubules, and thereby, can direct the centrosome position to apical regions, as seen during planar cell polarity establishment [18]. Next, major polarity regulators (PAR3, PAR6 and aPKC) and E-cadherin-mediated adherent junctions greatly influence the organization of apical non-centrosomal MTOCs in epithelial cells [13]. Notably, there are a few microtubules minus ends interacting proteins (-Tips), such as γ-tubulin, and calmodulin-regulated spectrin-associated proteins (CAMSAPs) members CAMSAP2, CAMSAP3, spectraplakin and ninein. CAMSAP and ninein represent two central mechanisms by which microtubule minus ends attach to cortical regions of the epithelial tissues in mammalian cells [19]. The re-localization of ninein to apical non-centrosomal MTOC sites helps to anchor microtubule minus ends at the apical membrane of polarizing epithelial cells [19].

Unlike epithelial cells, mesenchymal cells contain radial arrays of microtubules originating from centrosomal MTOCs in proximity to the nucleus and Golgi, which is the principal site of microtubule nucleation and growth (Figure 1C). Mesenchymal cells contain shorter and less numerous microtubules, with their centrosome localized at the cell center [20]. A typical mammalian cell centrosome contains a centrally localized centriole pair in perpendicular orientation with each other and is surrounded by a centrosomal matrix called pericentriolar material (PCM). The mass spectrometric analysis of purified centrosomes shows a large number of proteins that include, α-tubulin, β-tubulin, γ-tubulin, γ-tubulin complex components 1-6, centrin 2 and 3, AKAP450, pericentrin/kendrin, ninein, pericentriolar material 1 (PCM1), centriolin, CLIP-associating proteins CLASP1 and CLASP2, EB1, centractin, myomegalin, Cdk1, dynein intermediate chain, dynein light chain, p150Glued, dynactin and Hsp73 [21].

Overall, an off-centered non-centrosomal MTOC at apical regions and the maintenance of strong cell–cell adhesions via adherent and tight junctions in polarized epithelial cells impart tumor suppressive characteristics to epithelial cells. Conversely, the dissolution of cell–cell adhesion, the loss of apical-basolateral polarity and the gaining of front-back cell polarity, and the acquisition of adhesive and migratory characters all collectively facilitate cell adhesion, migration and cancer progression.

## 3. Reversion of MTOC from Non-Centrosomal to Centrosomal Positioning in Epithelial Cells Transitioning to Mesenchymal Cells

The dissolution of the apical-basolateral polarity axes of epithelial cells and establishment of a front-rear polarity axis are accomplished via the EMT process [22]. Along with the dissolution of cell–cell adherent junctions and cell polarity, the change from non-centrosomal to centrosomal MTOCs (centrosome repositioning) and the development of radially oriented arrays of microtubules from the centrosome at the cell center are prominent features observed in epithelial cells converting to mesenchymal cells [7]. Overall, this process is known as a polarity reversal and is an early and a key feature of the EMT [7]. The decades-old literature shows synchrony between centrosomal MTOC repositioning and epithelial cell migration, supporting the notion that internal polarity reversal is instrumental for cell migration [23]. The process of centrosomal MTOC repositioning concomitant with the changes in protein expression and induction of EMT is elegantly demonstrated in MCF10A and MDCK epithelial cells following TGF-β treatment in vitro (Figure 2) [7].

There is no precise mechanism explaining how the non-centrosomal MTOC in apical regions is translocated to a centrosomal localization as epithelial cells transition to mesenchymal cells [13,16]. Decreases in microtubule nucleation, polymerization and stabilization along adherent junctions and apical regions are possible mechanisms that may lead to the displacement of non-centrosomal MTOC to the centrosome adjacent to the nucleus and Golgi structures [6]. One of the critical factors is a decrease in the localization of cortical polarity components Par3 and podocalyxin in intercellular junctions and the reversion of the internal polarity axis [4]. Par3 is present along the epithelial cell junction and is capable of recruiting dynein, which pulls on microtubules, and thus, can determine the centrosome position during planar cell polarity establishment. The amount of polymerized tubulin also contributes to the transition from an off-centered microtubule network in epithelial cells to a centered conformation in mesenchymal cells [16].

Overall, the transition from epithelial cells to mesenchymal states is accompanied not only by cytoskeletal reorganization, but also the spatial reorganization of internal organelles and intracellular protein trafficking pathways [22]. Importantly, when the internal apical-basal polarity axis is lost by transitioning epithelial cells, the cell’s internal polarity axis guided by microtubules is directed toward the cell periphery and cell adhesion sites [7]. As microtubules serve as a track for internal trafficking, polarized delivery and targeting may become more efficient and streamlined due to polarized microtubules originating from centrosomal MTOC at the Golgi and nucleus towards the cell periphery guided by +Tips-directed kinesin motor proteins. This spatial reorganization may ultimately facilitate cell adhesion and migration. 

## 4. Microtubule Assembly and Its Inverse Relation with Focal Adhesion Assembly: Coordinated Interplay in Cell Adhesion and Cell Migration

Once apical-basolateral polarity is lost, transitioning cells establish front-rear cell polarity along with prominent centrosomal MTOC organization and radially orientated microtubules towards the cell periphery [24,25]. This conversion prepares the cells for adhesion and migration along the extracellular matrix proteins of interstitial tissues [26]. Intriguingly, the dynamics of microtubule polymerization and growth are inversely correlated with the assembly of the focal adhesions [10,27] (Figure 3). This necessitates the coordinated interplay between microtubule polymerization/depolymerization with focal adhesion assembly/disassembly to accomplish successful cell migration. The dynamic assembly and disassembly of focal adhesion at leading and trailing edges of migrating cells is one of the most extensively studied topic in cell adhesion and cell migration [28,29]. The assembly of focal adhesions at the leading edges is accompanied by the disassembly of the focal adhesions at the trailing edges, and these processes continue as the cells migrate [29]. 

Microtubule dynamics and orientation guide overall cell polarity and leading-edge protrusions [29]. Similarly, the adhesion complexes establish the leading edges of migrating cell fronts interacting with extracellular matrix proteins [30]. Thus, the establishment of front-rear cell polarity and cell migration is an integrated function of microtubule and cell adhesion complexes. Importantly, the dynamics of microtubule polymerization and depolymerization differ at the leading edge and cell rear and are carefully coordinated via adhesion assembly and disassembly [31]. The growing microtubule likely gains access to focal adhesion sites at the leading and trailing edges using actin stress fibers as tracks, with the aid of crosslink proteins MACF1/ACF7, KANK1 and Talin [32,33]. However, the direct interrogation of microtubule plus ends at focal adhesion sites remains unclear due to the presence of denser arrays of the actin cytoskeleton. Microtubules growth/retraction may require multiple repetitions at focal adhesion sites to accomplish focal adhesion turnover [29,31]. The process of microtubule disassembly or catastrophes at focal adhesions is triggered by a biochemical mechanism involving stathmin and catastrophe-inducing molecules, such as kinesin-13 family members MCAK [34]. Furthermore, microtubule growth and capture nearby focal adhesions, leading to the disassembly of focal adhesions via different mechanisms, including clathrin-mediated endocytosis, NBR1-mediated autophagy and the delivery of exocytic vesicles carrying matrix metalloproteases that sever integrin–ECM connection sites [35].

Another key mechanism by which microtubule polymerization and depolymerization associate with focal adhesions is by regulating Rho-GTPase signaling that controls actomyosin-based contractility [36]. Microtubule depolymerization (e.g., via a nocodazole treatment) results in an increase in RhoA activity, which correlates with increased focal adhesion assembly [36]. GEF-H1, a microtubule-associated guanine nucleotide exchange factor, activates RhoA upon release from microtubules, as microtubule depolymerization leads to the release of active GEF-H1, which is responsible for activating RhoA GTPase and focal adhesion assembly [36]. KANK family proteins connect microtubule tips with the cell–matrix adhesion site and focal adhesions with focal adhesion protein talin [37]. The disruption of microtubules or microtubules uncoupling from focal adhesions by manipulating KANKs triggers a massive assembly of actin filaments via myosin IIA and focal adhesions. Myosin IIA, a hexameric actin-binding protein, possesses ATPase activity at its head domain, enabling ATP hydrolysis and the generation of energy to move myosin along actin filaments [37]. The actin cross-linking and motor property of myosin IIA is essential for its regulation of cell adhesion and cell protrusions [37]. Notably, myosin IIA activation depends on Rho activation with RhoGEF GEF-H1, which is trapped in microtubule-KANK interaction sites. In contrast, microtubule regrowth rapidly stimulates Rac1 activation and lamellipodial ruffling, along with microtubule targeting of focal adhesions for disassembly [38]. Similarly, microtubule plus end tracking proteins, EB1, EB2 and EB3, guide growing microtubule plus ends towards focal adhesions for focal adhesion turnover at the leading edge of the migrating cells [39]. Other microtubule plus end-binding proteins, such as APC, MACF1/ACF7 and CLASPs, play roles in microtubule stabilization to control focal adhesions dynamics during cell migration [40]. 

Another key aspect of cell migration is the rate-limiting focal adhesion disassembly process. The implications of GTPase protein dynamin and vesicle coating protein clathrin suggest an endocytic process may be responsible for focal adhesion disassembly [10,41]. Focal adhesion disassembly involves the endocytosis of integrins adhesion molecules from focal adhesions [42]. Similarly, a significant proportion of vesicle fusion events occur in the vicinity of focal adhesions, and proteolytic enzymes in exocytic vehicles stimulate adhesion disassembly [43]. The proteolytic cleavage of focal adhesion proteins via intracellular calcium-dependent protease calpain2, which cleaves talin as calpain2, is present in the early endosomes in association with Rab5 [44]. Previously, we introduced the concept of PI4,5P_2_ generation in focal adhesion assembly through focal adhesion signaling via regulated interaction between talin, PIPKIγi2 and integrins [45,46]. Because most intracellular vesicle transport occurs along microtubules, microtubules likely serve as specific tracks for cargo transport from and/or to focal adhesions. It has been an established dogma that the coordinated secretion and recycling of integrin adhesion receptors are important for cell adhesion and migration, and integrins are internalized and shuttled via endosomal vesicles for exocytosis at the leading edge for the establishment of new focal adhesions [47]. The observations of PI4,5P_2_ generated via PIPKIγi2 and the spatial recruitment of exocyst complexes as PI4,5P_2_ effector molecules to drive polarized recruitment of integrin molecules at the leading edge of migrating cells align well with this established dogma [48]. 

Additionally, cell adhesiveness serves as a biophysical marker of metastatic potential. The intrinsic differences in the adhesion strength of cells within a population can act as markers of intra-tumoral heterogeneity and can be exploited to biophysically fractionate subpopulations [49]. A better understanding of how metastasizing cancer cell downregulate their cell adhesion during metastasis but upregulate cell adhesion at the time of intravasation or the establishment of early metastatic sites, may provide new avenues for therapeutic targeting of cell adhesion and migration in cancer.

## 5. Microtubules Link Cell Polarity and Spatial Phosphoinositide Signaling

Phosphoinositide lipid molecules regulate epithelial cell polarity and cell adhesion and migration via different mechanisms. For example, PI4,5P_2_ phosphoinositide molecules regulate clathrin adaptor proteins for the polarized delivery of E-cadherin molecules and epithelial morphogenesis [50]. Epithelial cells maintain the segregation of phosphoinositide molecules, PI4,5P_2_ at the apical plasma membrane domain and PI3,4,5P_3_ at the basolateral plasma membrane domain [51,52]. The spatial segregation of PI4,5P_2_ and PI3,4,5P_3_ is critical for maintaining the epithelial cell polarity as changes in the distribution of PI4,5P_2_ to basolateral domains redirect apical proteins to basolateral domains, and vice versa. PI4,5P_2_ at an apical domain recruits Annexin 2 [51]. PTEN, which dephosphorylates PI3,4,5P_3_ to PI4,5P_2_, maintains low levels of PI3,4,5P_3_ at the apical membrane. Type I phosphatidylinositol 4-phosphate 5-kinase (PIP5K) generates PI4,5P_2,_ which sustains microtubule organization to establish polarized transport and cell polarity in Drosophila oocytes [53]. Additionally, a recent study reported that another phosphoinositide molecule, PI3,4P_2_, localizes at the apical surface and Rab11α-positive apical recycling endosomes [54]. The perturbation of PI3,4P_2_ generation impairs the delivery of vehicles destined for apical regions. However, the precise role and involvement of the microtubule as a track for intracellular trafficking in segregating distinct pools of phosphoinositides remains unclear. 

PI3,4,5P_3_ generation and its downstream signaling is portrayed as a universal signal for cell survival/growth during epithelial morphogenesis [55]. Although studies show an intimate association of PI3-kinase (PI3K) with microtubules via direct interaction with tubulin or indirectly via microtubule-associated protein 4 (MAP4) [56], the significance of PI3K in cell polarity, cell adhesion and cell migration remains to be defined. The homophilic cell–cell adhesion mediated via E-cadherin molecules is responsible for driving PI3K activation and PI3,4,5P_3_ generation at the basolateral domains of epithelial cells for epithelial morphogenesis and cell polarity [55,57,58] (Figure 4). Although agonist-stimulated receptor tyrosine kinase activation and PI3,4,5P_3_ generation are transient and rapid, E-cadherin-mediated homophilic cell–cell adhesion and PI3K recruitment and PI3,4,5P_3_ generation at the basolateral domain of the polarized epithelial cells may be more constitutive [59,60]. Another unresolved question is whether the orientation of microtubules and the motor proteins employed, kinesin vs. dynein in epithelial cells and mesenchymal cells, respectively, have any regulatory role in the spatial generation of the PI3,4,5P_3_ lipid messenger and Akt activation? The investigation of the spatial generation of PI3,4,5P_3_ and Akt activation in cell types with or without intact E-cadherin-mediated adherent junctions may shed light on defining spatial PI3,4,5P_3_ and Akt activation in addition to interrogating the role of microtubules and microtubule-associated proteins, including MAP4, during these events (Figure 4) [56].

An important link between phosphoinositides and directional cell migration is the spatial generation of PI3,4,5P_3_ at the leading edge and the polarized recruitment of signaling molecules required for cell migration [61,62]. However, these processes have been largely studies in rapidly migrating leukocytes. Similarly, PI3,4P_2_ and PI3,4,5P_3_ generation is required for invadopodium formation and invasive cell migration [63,64].

## 6. Therapeutic Targeting of Microtubules in Cancer and the Effects on Focal Adhesion Signaling

Microtubule targeting drugs represent one of the most successful first-line anti-cancer therapies in clinics since the advent of modern chemotherapy [11]. These drugs target the microtubule mitotic spindles required for chromosome segregation, thereby leading to mitotic arrest and cell death [11]. Notably, microtubule-targeted anti-cancer drugs also impair cell adhesions and migration [11,65]. However, like many other anti-cancer drugs, the long-term use of microtubule-targeted anti-cancer drugs inevitably results in the development of drug resistance [66]. Elevated efflux pumps due to the induced expression of different ATP-binding cassette transporter systems and drug efflux from cancer cells are well-recognized mechanisms for the development of the acquired resistance [66]. Additionally, the increased expression of microtubule-associated proteins in cancer can mask or modify the susceptibility of microtubules caused by microtubule-targeted anti-cancer drugs [67,68]. 

Moreover, microtubule depolymerization promote focal adhesion assembly [69]. Indeed, microtubule depolymerizing anti-cancer drugs (e.g., vincristine) increases the focal adhesion kinase activity and focal adhesion signaling rates [70]. The disruption of microtubules results in the enlargement of adhesion complex size, adhesion signaling due to increased RhoA-stimulated actomyosin contractility and the inhibition of adhesion complex turnover [71]. Increased levels of RhoA GTPase and its associated signaling may antagonize the effect of microtubule-targeted drugs in cancer therapy. It is unclear whether the observed enhanced focal adhesion signaling plays a role in the development of drug resistance to microtubule-targeted anti-cancer drugs.

Although the therapeutic targeting of microtubules and its effect on cell adhesion and migration in cancer seems perplexing, the altered expression of the tubulin isotypes in microtubules is associated with aggressiveness among different cancer types, altered sensitivity to chemotherapy and the development of drug resistance [72]. Furthermore, the post-translational modification of tubulin (e.g., phosphorylation, acetylation, methylation, palmitoylation, ubiquitylation and polyamination) modifies an intrinsic property of microtubules and microtubule functions [73]. 

## 7. Conclusions and Future Direction

In this Special Issue of “Molecular Mechanism Underlying Cell Adhesion and Migration”, we have highlighted cell adhesion and migration from the perspectives of microtubule reorganization, as epithelial cells become transformed and transition to adhesive and migratory mesenchymal phenotypes. Though cell adhesion and cell migration have been considerably studied in the context of the actin-based cytoskeleton, the functional contribution of microtubules has been increasingly recognized and explored in the context of cell adhesion and migration [36]. Overall, microtubules play important roles in maintaining the shape of the cells, but also function in intracellular vesicle trafficking, organelles repositioning and cell signaling [36]. Conspicuously, the microtubule cytoskeleton undergoes dramatic spatial and temporal reorganization, including the originating or nucleation sites as epithelial cells transform into adhesive and migrating mesenchymal cells [7]. Yet, our precise and in-depth understanding of how these dramatic changes in nucleation or originating sites of microtubules, its overall orientation inside the cells, and the resulting effect on cell adhesion and migration remains incomplete. Hyperactive MTOC and enhanced microtubule nucleation at the centrosome are increasingly recognized as hallmarks of cancer [21,74]. Clearly, a better molecular understanding of how microtubule organization impacts cell adhesion and migrations may open new avenues for cancer therapy. Traditionally, cell adhesion and migration have been largely studied using 2D culture conditions. Our understanding of the different events choreographed in cell migration, such as the establishment of front-rear cell polarity, microtubule reorganization, leading edge extensions, assembly, and disassembly of focal adhesions at leading and trailing edges, is largely based on studies performed in 2D. Similarly, the identification of the role of different molecules in cell adhesion and migration has largely relied on 2D assays. As the 3D in vivo environment is fundamentally different from the 2D environment, the discovery of the role of different molecules in cell adhesion and migration in the 3D microenvironment that more accurately recapitulates the in vivo environment will provide new insights about these key cellular events that underlie cancer [75].

## Figures and Tables

**Figure 1 biomolecules-13-01430-f001:**
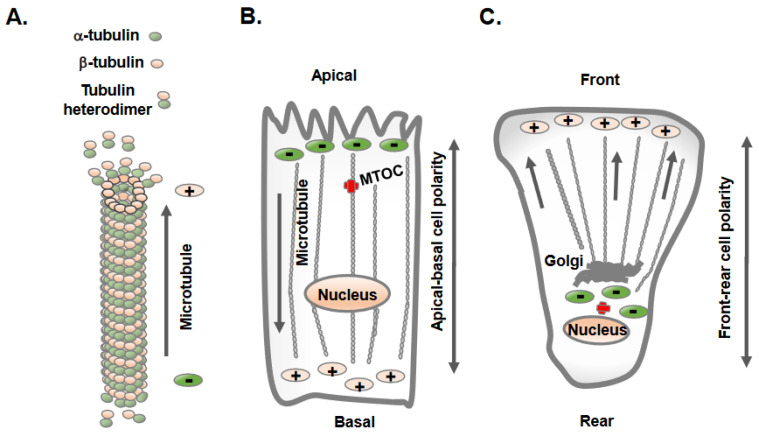
Epithelial and mesenchymal cells display distinct orientation and organization of microtubules. (**A**) Microtubules are hollow tubes composed of 13-individual polymers of tubulin; each tubulin molecule is heterodimer of α-tubulin and β-tubulin. The polymerization and growth of microtubules often takes place at the plus end with β-tubulin, whereas the minus end contains α-tubulin. (**B**) Epithelial cells maintain apical-basolateral cell polarity, with the microtubule minus ends located at apical regions and plus ends located at the basolateral regions. (**C**) Mesenchymal cells acquire front-rear cell polarity and microtubules extend to the cell periphery from the cell center.

**Figure 2 biomolecules-13-01430-f002:**
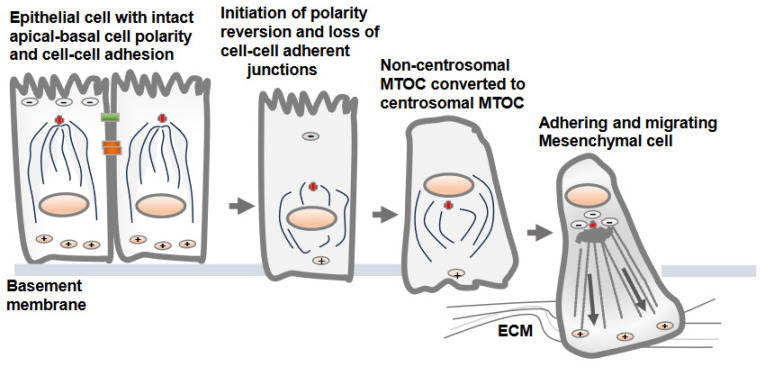
Non-centrosomal MTOC in epithelial cells adopts a centrosomal MTOC position as epithelial cells transition to mesenchymal cells. There is no well-defined mechanism by which non-centrosomal MTOC is reverted to centrosomal MTOC as epithelial cells convert into mesenchymal cells. The development of centrosomal MTOC from non-centrosomal MTOC as epithelial cell convert into mesenchymal cells is referred as cell polarity reversion. Upon loss of the cell–cell adherent junctions and apical-basolateral cell polarity, the MTOC is gradually developed at the cell center, from which polarized microtubules radiate towards the cell periphery. This is accompanied by gain of adhesive and migratory traits via transitioning cells.

**Figure 3 biomolecules-13-01430-f003:**
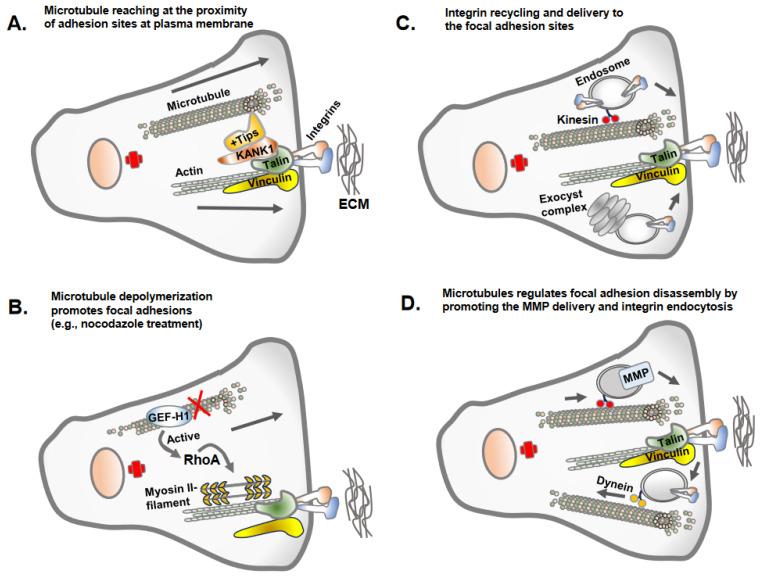
Microtubule regulation of focal adhesions. There are different mechanisms by which the microtubule regulates focal adhesions. (**A**) Microtubules often do not reach the site of focal adhesions due to the dense actin cytoskeleton but make indirect contact via KANK1 or +Tips proteins, stabilizing the plus ends of the microtubules. (**B**) The key mechanism by which microtubule polymerization or growth inversely regulate focal adhesions is due to the activity of GEF-H1, the guanine exchange factor H1 for RhoA GTPases. GEF-H1 bound to microtubules is in an inactivated state. GEF-H1 is activated once released from depolymerized microtubules (e.g., nocodazole treatment of cells) and activates RhoA GTPase, which promotes actin polymerization and contractility and increased focal adhesion assembly. (**C**) Microtubules as a track for endosomal trafficking promote the delivery of the newly synthesized or recycling integrins to newly forming focal adhesion sites. An evolutionary conserved vesicle trafficking complex, exocyst, in association with microtubules may also deliver integrin molecules to focal adhesion sites. (**D**) Microtubules serve as a track for endosomal trafficking of matrix metalloprotease (e.g., MT1-MMP) at the site of focal adhesion, leading to the disassembly of the focal adhesions. The integrins once internalized from adhesion sites undergo endosomal trafficking along microtubules tracks.

**Figure 4 biomolecules-13-01430-f004:**
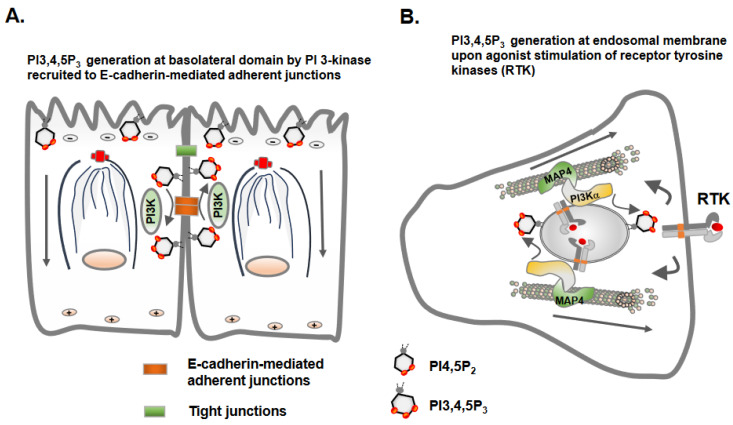
Spatial generation and localization of phosphoinositide signaling molecules in epithelial and mesenchymal cells. (**A**) The spatial localization of phosphatidylinositol-4,5-bisphosphate (PI4,5P_2_) at apical plasma membrane regions and phosphatidylinositol-3,4,5-trisphosphate (PI3,4,5P_3_) at basolateral regions are crucial for maintaining the apical-basolateral cell polarity of epithelial cells. PI3,4,5P_3_ is likely generated constitutively at baso-lateral plasma membrane solely by PI3K recruited to E-cadherin molecules at the adherent junctions. (**B**) Mesenchymal cells in which apical-basolateral cell polarity is lost may generate PI3,4,5P_3_ predominantly at endosomal and plasma membranes downstream of activated receptor tyrosine kinases.

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
