# Peer review of "Regulation of Cell Adhesion and Migration via Microtubule Cytoskeleton Organization, Cell Polarity, and Phosphoinositide Signaling"

_biomolecules, 2023, doi:10.3390/biom13101430_

Round 1
Reviewer 1 Report
This is an interesting review that overviews microtubule reorganization in cancer. Typically reviews focus on the actin cytoskeleton, so this review represents a different angle that is useful to the field.
I have a few comments:
1. From the abstract and title it appears as if the review will discuss all cytoskeleton, however the focus here is entirely on microtubules. This is appropriate, since cytoskeleton is a huge field that would be difficult to overview in one article, but the introductory sections of the review, including abstract and title, need to be edited to accurately reflect the focus. In fact, this clarification would make the review more interesting and different.
2. The authors mention the themed collection of the issue several times in the review. This information is irrelevant to the review and should be removed.
3. The processes of microtubule reorganization discussed here are not specific to cancer and EMT transition. This needs to be correctly reflected, and the global nature of these processes should be discussed. Any information specific to cancer should be clearly identified, if present. In fact, such information, if available, merits a separate section in the review.
4. One big aspect of microtubules in cancer relates to tubulin isoform specificity. While the underlying mechanisms are not presently understood, it would be nice to see this discussed in the review. I also felt that some information on posttranslational modifications of microtubule proteins would be interesting to add.
Author Response
We would like to thank reviewers for their insightful comments and suggestions that have greatly helped us to strengthen our revised manuscript. Please see point-by-pint response to all the comments/suggestions.
Reviewer 1: This is an interesting review that overviews microtubule reorganization in cancer. Typically reviews focus on the actin cytoskeleton, so this review represents a different angle that is useful to the field. I have a few comments:
Answer: We would like to thank the reviewer for these encouraging comments.
1. From the abstract and title it appears as if the review will discuss all cytoskeleton, however the focus here is entirely on microtubules. This is appropriate, since cytoskeleton is a huge field that would be difficult to overview in one article, but the introductory sections of the review, including abstract and title, need to be edited to accurately reflect the focus. In fact, this clarification would make the review more interesting and different.
Answer: We agree with reviewer’s comments/suggestions. Please see edited section in the abstract line 16.
2. The authors mention the themed collection of the issue several times in the review. This information is irrelevant to the review and should be removed.
Answer: We agree with reviewer’s comments/suggestions. We have removed the mentioning of “themed collection of the issue several times” in our revised manuscript.
3. The processes of microtubule reorganization discussed here are not specific to cancer and EMT transition. This needs to be correctly reflected, and the global nature of these processes should be discussed. Any information specific to cancer should be clearly identified if present. In fact, such information, if available, merits a separate section in the review.
Answer: We agree with reviewer’s comments/suggestions that the processes of microtubule reorganization discussed here are not specific to cancer and EMT transition. We have clarified this in the introduction section and added two additional references on page 2 line 48.
4. One big aspect of microtubules in cancer relates to tubulin isoform specificity. While the underlying mechanisms are not presently understood, it would be nice to see this discussed in the review. I also felt that some information on posttranslational modifications of microtubule proteins would be interesting to add.
Answer: We agree with reviewer’s comments/suggestions. We have included references and a brief discussion on page 10 line 358.
Reviewer 2 Report
Thapa et al. provide a state-of-the art, clearly written review about the functions of microtubules in cell polarity. They start by talking about the poorly understood switch from epithelial, non-centrosomal orientation of microtubules directed at basolateral membranes towards a mesenchymal perinuclear organization whereby microtubules orient vesicular traffic to support the front of migrating cells. They then move on to analyze the inverse relationship between microtubules dynamics and focal adhesions; the poorly investigated link between phosphoinositide signaling and microtubules; and a short comment about microtubule-targeting agents in cancer therapy.
The choice of beginning with the change in MTOC positioning during epithelial to mesenchymal transition was well pondered as everything falls in the right place afterwards, making the reading extremely pleasant and clear.
Overall, my opinion on this review is very positive: the topic is of interest, all aspects are presented in a balanced manner, and reading makes one willing to start working on these topics. Hence, I wish to congratulate to the authors for their nice work.
I have two minor suggestions, but it is up to the authors to decide whether they want to follow them or not:
- Briefly mentioning the studies from the Bershadsky lab about how microtubules, FAs and GEF-H1 are linked (https://doi.org/10.1038/s41563-019-0371-y and https://doi.org/10.1101/2023.04.17.535593) could further improve the already excellent section on focal adhesions and microtubules.
- Figure 4 is very pixelated. Maybe it is just an uploading problem, but it would be a pity to have low quality images on such a beautiful text
Author Response
We would like to thank reviewers for their insightful comments and suggestions that have greatly helped us to strengthen our revised manuscript. Please see point-by-pint response to all the comments/suggestions.
Reviewer 2: Thapa et al. provide a state-of-the art, clearly written review about the functions of microtubules in cell polarity. They start by talking about the poorly understood switch from epithelial, non centrosomal orientation of microtubules directed at basolateral membranes towards a mesenchymal perinuclear organization whereby microtubules orient vesicular traffic to support the front of migrating cells. They then move on to analyze the inverse relationship between microtubules dynamics and focal adhesions; the poorly investigated link between phosphoinositide signaling and microtubules; and a short comment about microtubule targeting agents in cancer therapy. The choice of beginning with the change in MTOC positioning during epithelial to mesenchymal transition was well pondered as everything falls in the right place afterwards, making the reading extremely pleasant and clear. Overall, my opinion on this review is very positive: the topic is of interest, all aspects are presented in a balanced manner, and reading makes one willing to start working on these topics. Hence, I wish to congratulate the authors for their nice work. I have two minor suggestions, but it is up to the authors to decide whether they want to follow them or not:
Answer: We would like to thank the reviewer for these encouraging comments.
1. Briefly mentioning the studies from the Bershadsky lab about how microtubules, FAs and GEF-H1 (https://doi.org/10.1038/s41563-019-0371-y are linked and https://doi.org/10.1101/2023.04.17.535593) could further improve the already excellent section on focal adhesions and microtubules.
Answer: We have cited indicated references and included the concise description on page 7 line 241.
2. Figure 4 is very pixelated. Maybe it is just an uploading problem, but it would be a pity to have low quality images on such a beautiful text.
Answer: All the original figures in power point are available and we will upload individually if needed.